# Testing for reviewer anchoring in peer review: A randomized controlled trial

**Ryan Liu**◉*, **Steven Jecmen, Vincent Conitzer, Fei Fang, Nihar B. Shah**◉

School of Computer Science, Carnegie Mellon University, Pittsburgh, Pennsylvania, United States of America

* ryanliu@princeton.edu

## Abstract

### Objective

Peer review frequently follows a process where reviewers first provide initial reviews, authors respond to these reviews, then reviewers update their reviews based on the authors' response. There is mixed evidence regarding whether this process is useful, including frequent anecdotal complaints that reviewers insufficiently update their scores. In this study, we aim to investigate whether reviewers *anchor* to their original scores when updating their reviews, which serves as a potential explanation for the lack of updates in reviewer scores.

### Design

We design a novel randomized controlled trial to test if reviewers exhibit anchoring. In the experimental condition, participants initially see a flawed version of a paper that is corrected after they submit their initial review, while in the control condition, participants only see the correct version. We take various measures to ensure that in the absence of anchoring, reviewers in the experimental group should revise their scores to be identically distributed to the scores from the control group. Furthermore, we construct the reviewed paper to maximize the difference between the flawed and corrected versions, and employ deception to hide the true experiment purpose.

### Results

Our randomized controlled trial consists of 108 researchers as participants. First, we find that our intervention was successful at creating a difference in perceived paper quality between the flawed and corrected versions: Using a permutation test with the Mann-Whitney $U$ statistic, we find that the experimental group's initial scores are lower than the control group's scores in both the Evaluation category (Vargha-Delaney $A = 0.64$, $p = 0.0096$) and Overall score ($A = 0.59$, $p = 0.058$). Next, we test for anchoring by comparing the experimental group's revised scores with the control group's scores. We find no significant evidence of anchoring in either the Overall ($A = 0.50$, $p = 0.61$) or Evaluation category ($A = 0.49$, $p = 0.61$). The Mann-Whitney $U$ represents the number of individual pairwise comparisons across groups in which the value from the specified group is stochastically greater, while the Vargha-Delaney $A$ is the normalized version in [0, 1].

**Data Availability Statement:** All code and data files are available on GitHub at the following repository: https://github.com/theryanl/ReviewerAnchoring.

**Funding:** NS received the National Science Foundation CAREER Award 1942124 (https://www.

nsf.gov/). The funders had no role in study design, data collection and analysis, decision to publish, or preparation of the manuscript. NS and FF received the National Science Foundation Division of Information and Intelligent Systems 2200410 (https://new.nsf.gov/cise/iis). The funders had no role in study design, data collection and analysis, decision to publish, or preparation of the manuscript.

**Competing interests:** The authors have declared that no competing interests exist.

# 1 Introduction

Peer review is the primary method for systematically evaluating scientific research. Many peer-review processes involve reviewers submitting an initial review, following which they may be presented with additional information. This additional information frequently takes the form of a response from the authors. The reviewers are then requested to read the response and adapt their stated opinions and evaluations accordingly. In this work, we put this potential change under the microscope, investigating whether reviewers *anchor* to their original opinions. For concreteness, we instantiate our study in the setting of conference peer review, a large human-centric system that has been widely adopted in computer science academia. (In computer science, leading conferences are typically rated at least on par with leading journals, with full paper submissions, competitive acceptance rates from 15–25%, and are often terminal venues for publication.) Across conference peer review, the author response mechanism is termed the "rebuttal stage", placed between the initial reviews and final review score decisions and are an opportunity for the author(s) to provide additional information or arguments in response to the initial reviews. (Depending on the specific review setting, there may also be alternative forms of information made available to the reviewer, such as the evaluations of other reviewers. In this work, we focus on author rebuttals due to its widespread use and frequently-raised questions about its efficacy.) In computer science conferences, rebuttal stages are a widely adopted practice, with a large number of recent conferences having instituted such periods [1, 2].

Despite its pervasiveness, there is so far mixed evidence regarding the usefulness of rebuttals. A program chair of the NAACL 2013 conference described the rebuttal phase as "useless, except insofar as it can be cathartic to authors and thereby provide some small psychological benefit" [3]. A study on the NeurIPS 2016 conference found that only 4180 of 12154 (34.4%) reviews had reviewers participate in the discussion after the rebuttal, and only 1193 (9.8%) of reviews subsequently changed in score [4]. Furthermore, adjustments in reviewer scores do not necessarily affect paper decisions—in the ACL 2018 conference, 13% of reviewer scores changed after rebuttals, but the amount of papers whose acceptances were likely affected was only 6.6% [1]. In addition, authors from various conferences have shared vast amounts of anecdotes on social media regarding the limited impact of their rebuttal statements on reviewer evaluations, including cases where they had written a strong rebuttal but reviewers did not respond to it in a fair and reasonable way [5–7]. Rogers and Augenstein [8] find that in the natural language processing community, Twitter posts drastically spike both during the rebuttal phase and at acceptance notifications (corresponding to when authors create their rebuttals and when they see the results after rebuttals), with these tweets often including bitter complaints and reform suggestions.

One potential explanation behind the limited effect of the rebuttal stage on overall acceptances is that, due to *anchoring*, reviewers are simply not changing their scores as much as they should. Anchoring [9] is formally defined as the bias where people who make an estimate by starting from an initial value and then adjusting it to yield their answer typically make insufficiently small adjustments. Anchoring effects have been found in many applications, including responses to factual questions, probability estimates, legal judgments, purchasing decisions, future forecasting, negotiation resolutions, and judgements of self-efficacy [10–15]. However, despite the high stakes of peer review, anchoring has not yet been studied in the context of conferences and the rebuttal process.

## 1.1 Research question

In this paper, we test for the existence of anchoring in reviewers to verify whether reviewers are biased in a systematic manner. Our research question compares the following two scenarios in which a reviewer evaluates an academic paper.

- **Scenario A**: The reviewer evaluates the paper's quality and provides a set of numeric scores (termed initial scores). The reviewer is then presented with additional evidence proving that their initial evaluation was mistaken. Subsequently, the reviewer optionally adjusts their previous scores to new values (termed revised scores).

- **Scenario B**: The reviewer is simultaneously presented with the same paper and the additional evidence from the previous scenario. They then provide a numeric evaluation of the paper's quality (termed control scores).

Here, scenario A is a situation that may occur in a typical rebuttal process. Scenario B is a counterfactual where the additional evidence of scenario A is incorporated into the paper and presented to the reviewer during their initial reading of the paper. If anchoring is present in the rebuttal process, reviewers' revised scores in scenario A would remain closer to their lower initial scores, and not be identical to the scores they would have given if they had been in scenario B. In aggregate, this would lead to a muted change in acceptances and a less effective rebuttal process.

Altogether, we study the following research question: *Are the revised scores given by reviewers when placed in scenario A lower than the control scores that those reviewers would have given if they had been placed in scenario B?*

We hypothesize that, in line with the existing literature on anchoring, reviewers in scenario A will anchor to their initial review scores, causing their revised scores to be lower than the control scores they would have given if they had been in scenario B.

## 1.2 Our contributions

To answer the research question, we designed and conducted a study to analyze the reviewer anchoring effect.

1. We recruited 108 participants who have recently published in a computer science-related field and are currently pursuing or have completed their PhD, and randomly assigned them to the control or experimental group. Each participant was placed in the role of a reviewer in a mock conference setting and was asked to review one paper.

2. We constructed a fake paper for participants to review, and showed different versions of the paper to the different groups. The control group was given a paper with an animated GIF graphic (shown in Fig 1A) that contains the main evaluation results of the paper's proposed framework, while the experimental group was instead given a frozen frame of the GIF (Fig 1B) that showed a much weaker result. After experimental group participants completed their review, they were deceived that the GIF was frozen as the result of a technical error, and were shown the proper animated GIF, upon which they were given the opportunity to revise their scores. Our experiment was carefully designed to avoid several confounders and challenges in simulating an anchoring effect under the rebuttal setting, which we detail in Section 3.1.1.

3. For the paper, each reviewer was asked to provide an overall score, five category scores, and text comments justifying each category score. We collected this data once from the control group (control scores) and twice from the experimental group (initial and revised scores).

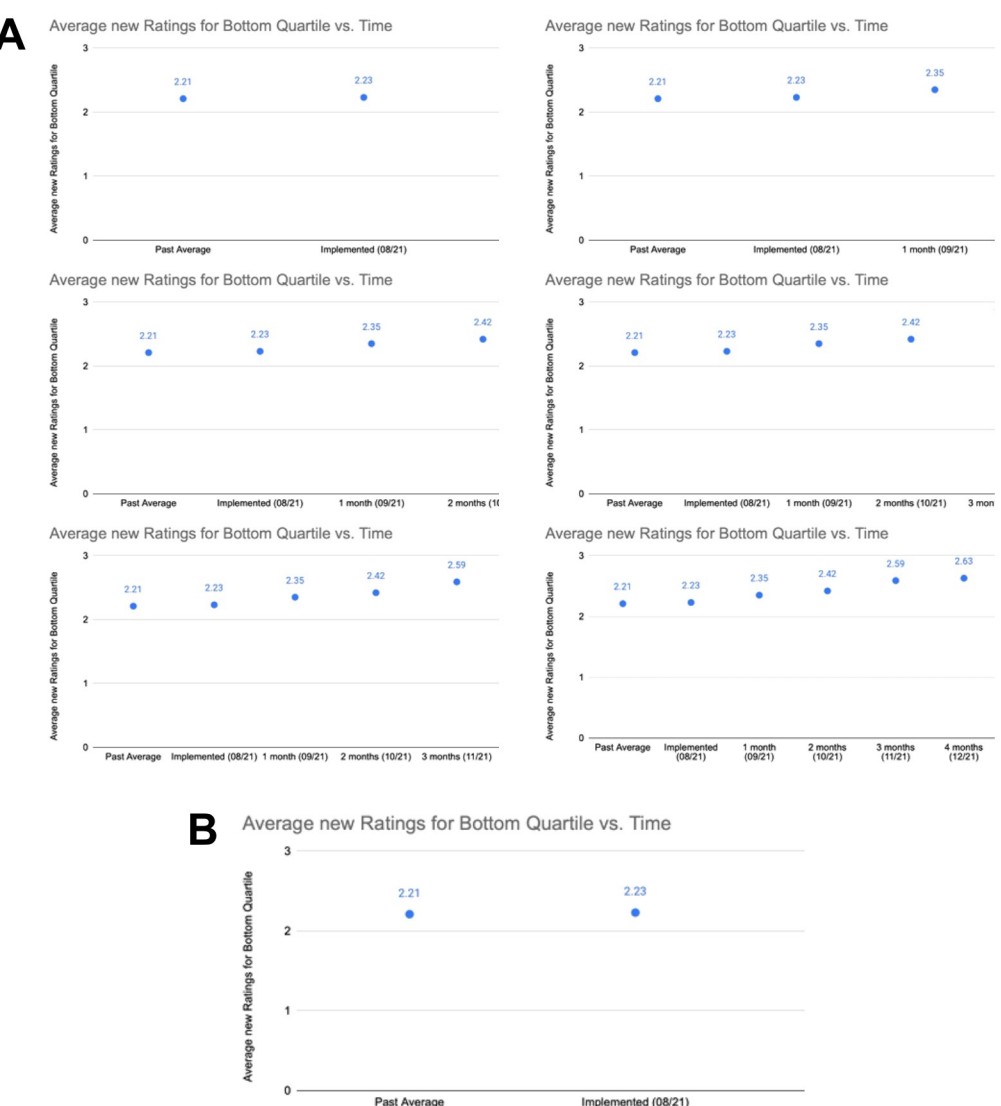

**Fig 1. The evaluation results used in the fake paper.** A: Chronological frames (from left to right) demonstrating the animated result GIF. B: Frozen GIF initially shown to the experimental group. The animation compresses the existing data points to the left, introducing more data points to the right in chronological fashion. Larger improvements on the y-axis correspond to a better evaluation result for the paper's method. The baseline is the leftmost point in all frames, 2.21 on a 1–5 scale. In the frozen figure (B), the rightmost point is 2.23, representing an improvement of 0.02 ($< 2\%$). In the animated figure (A), the rightmost point is 2.63, representing an improvement of 0.4 ($> 33\%$). The animated figure can be viewed at https://github.com/theryanl/ReviewerAnchoring/blob/main/fake_paper/images/animated_plot.gif.

We also collected participant data such as self-reported confidence, PhD year and institution. The de-identified data and analysis code are available on GitHub at https://github.com/theryanl/ReviewerAnchoring.

4. In our analysis, we first checked whether our GIF manipulation created a difference in reviewer ratings. We compared the initial scores and control scores, in both the Overall rating and the Evaluation category (which directly corresponds to the aspect of the paper we manipulated). We conducted a one-sided permutation test with the Mann-Whitney *U*

statistic and measured the effect size in terms of the Vargha-Delaney $A$ [16], representing the probability that a randomly-chosen control score is greater than a randomly-chosen experimental score (breaking ties uniformly at random). We found that the initial scores were lower than the control scores in both the Evaluation category (effect size $A = 0.64$, $p = 0.0096$) and Overall scores (effect size $A = 0.59$, $p = 0.058$), with moderate effect sizes. Thus, our experimental setup successfully introduced a difference in paper quality that enabled our test for anchoring.

To test for the anchoring effect, we compared the revised scores with the control scores using a one-sided permutation test with the Mann-Whitney $U$ statistic. We did not find significant evidence of reviewer anchoring in either the Overall scores (effect size $A = 0.50$, $p = 0.61$) or Evaluation category scores (effect size $A = 0.49$, $p = 0.61$).

Although our experiment imitates a specific rebuttal process in conference peer review, we take the first step in extending the literature on anchoring bias to the academic peer review setting, where individual expertise and knowledge may interact differently with human biases. To our knowledge, this is the first randomized controlled trial on anchoring in peer review. Our work could potentially be informative for similar academic settings, such as anchoring in reviewer discussion phases and longer-term author feedback processes.

In the following sections, we give a more comprehensive view on our work. In Section 2, we give context to how our work fits into the broader literature on conference peer review and human biases. In Section 3, we detail our experimental design, data collection, and analysis methods, and describe the various challenges that our design addresses. In Section 4, we report the results for our analyses. In Section 5, we present the takeaways and discuss the limitations for our current work, and propose directions for future research.

## 2 Related work

In this section, we give a brief outline of the work done in several areas: Research done to improve the conference peer review process, studies on cognitive biases in academic reviewers, sources relating to the rebuttal process in particular, and psychology literature regarding the anchoring bias.

### 2.1 Conference peer review

Conference peer review has been an increasingly active area of research due to the need for automated and scalable solutions, especially in the field of computer science [17]. Work has focused on improving the quality of reviewer assignments [18–22], providing robustness to malicious behavior [23–25], and addressing issues of miscalibration [26, 27] and subjectivity [28] between reviewers. Of particular relevance is the literature investigating cognitive biases in reviewers. These include studies on confirmation bias [29], commensuration bias [30], the effects of revealing author identities to reviewers [31–34], reviewer herding [35], resubmission bias [36], citation bias [37], and others [38]. Other works propose methodology for detecting such biases [33, 39].

Research has also focused on the reviewer discussion phase of peer review, which has some similarities to the rebuttal process we study. Most peer review processes include a reviewer discussion phase after initial reviews are submitted, where reviewers can read and respond to each others' reviews. Similar to the author rebuttal process, reviewers are allowed to update their reviews after receiving this new information. Several studies [40–42] on reviewer discussions in grant proposal reviews have found that disagreement between reviewers greatly

decreases after discussion, indicating that reviewers do update their scores to reach consensus. In one experiment [43], 47% of reviewers updated their review scores after being shown scores from other fictitious reviewers. Authors of [35] conducted a randomized controlled trial in the ICML conference to investigate the existence of herding in reviewer discussions, but found no evidence for this effect. While these studies provide insights into how reviewers update their opinions, the present work focuses specifically on anchoring in the rebuttal process.

## 2.2 Rebuttal processes

Many conference organizers have analyzed the rebuttal process within their own conferences, and the common finding is that rebuttals only make a meaningful difference to a small fraction of submissions. Out of the 2273 rebuttals at CHI 2020, 931 (41%) did not result in a mean score change, 183 (8%) resulted in an absolute mean score change of 0.5 or more, and only 6 (0.3%) saw the mean score change by 1 or more [44]. In ICML 2020, only 43% of reviewers updated their review in response to author rebuttals [45]. In ACL 2018, 13% of review scores changed after rebuttals, affecting 26.9% of all papers, but only 6.6% of papers were likely impacted in terms of acceptance [1]. At the same venue, though author responses had a marginal but statistically significant influence on final scores, a reviewer's final score was largely determined by their initial score and distances to scores given by other reviewers [46].

Despite these statistics, there is overwhelming support for the rebuttal stage from the research community. A set of surveys from PLDI 2015 [47] found that authors strongly value the rebuttal process; 96% of authors agreed (with 88% strongly agreeing) that they should be provided the opportunity to rebut reviews. Meanwhile, only 44% of authors agreed to the statement that their reviews were constructive and professional, and only 41% of authors agreed that their reviewers had sufficient expertise. Rogers and Augenstein [8] found that both the rebuttal stage and the acceptance results after rebuttals yield large increases in the number of tweets in the NLP research community, often including bitter complaints and reform suggestions. In an author survey for IEEE S&P 2017 [48], which did not have a rebuttal phase, approximately 30% of less experienced and 20% of experienced authors felt like they could have convinced their reviewers to accept their paper if they were given an opportunity for a rebuttal. Together, these results send the message that authors are often dissatisfied with their reviews, and that they strongly value the rebuttal mechanism as a method to address bad reviewing.

## 2.3 Anchoring bias

Anchoring (more specifically, the anchor-and-adjust hypothesis) was initially described by Tversky and Kahneman [9], who defined it as the effect where people who make an estimate by starting from an initial value and then adjusting it to yield their answer typically make insufficiently small adjustments. The initial value can be irrelevant to the question asked, and can also be a partial computation by the person themselves. One basis to interpret this behavior [49] is to view it as a cognitive shortcut: to reduce the mental strain of incorporating new evidence, individuals take their starting estimate and integrate new information in a naive, insufficient way. The anchoring effect has been shown to be present in a variety of domains and applications [10–15]. However, to our knowledge, our study is the first randomized controlled trial to analyze whether reviewers exhibit anchoring behaviors in peer review.

## 3 Methods

In this section, we describe the experiment we conducted and the analysis methods we employed to investigate the research question specified in Section 1. We first define the

experimental procedure along with associated justifications, and then describe participant recruitment and data collection. Lastly, we describe the analysis we performed on the data. Our research question and study design were pre-registered at https://aspredicted.org/W94_GD3. This experiment was approved by the Carnegie Mellon University Institutional Review Board (Federalwide Assurance No: FWA00004206, IRB Registration No: IRB00000603).

## 3.1 Experiment design

In this subsection, we first describe the challenges inherent to this problem setting before concretely defining the experimental procedure. We then articulate how our key design choices allow us to surmount these challenges.

**3.1.1 Challenges for the design.** First and foremost, our hypothesis cannot be tested with an experiment in a real conference environment as it is impossible to control the quality of papers and the strength of rebuttals. Thus, we carefully designed an environment for our experiment that simulates a real conference. In designing our experiment and simulated environment, we address four main challenges:

1. **Clarity and objectivity of the quality of rebuttal**. In a real conference environment, the impact of a rebuttal argument on its paper's quality is often subjective. This makes it hard to distinguish between an anchoring effect and a genuine belief that the rebuttal was weak. In our experiment, the rebuttal must clearly and objectively improve the quality of the paper. Furthermore, the participants chosen need to be able to detect this improvement. Lastly, the rebuttal should be meaningful no matter what participants write in their initial review.

2. **Addressing "author mistake" confounder**. When reviewing, reviewers find and comment about mistakes in the submission that are important to the quality of the paper. Even when authors address these mistakes, if these mistakes were influential enough in the first place, reviewers may choose to take them into account and penalize the authors by giving a lower score. In this study, we explicitly choose to focus on anchoring with respect to reviewer opinions about the paper itself and not their opinions about the authors. As such, we label this phenomenon as the *author mistake* confounder, and consider it to be distinct from the anchoring effect in our research question. In our experiment, we want to account for this confounder, and separate its effects from the anchoring bias.

3. **Equality of the experimental and control experiences**. In the experiment, we want to compare between an experimental group, which sees a rebuttal and adjusts their scores, and a control group, which gives the ground truth scores that the experimental group should ideally adjust to. In order to make a meaningful comparison between groups, we want the control group's paper to be equivalent to the experimental group's initial paper combined with the rebuttal. In the traditional conference form, this is paradoxical to recreate; rebuttals are constructed to directly address initial reviews, but the control group cannot give initial reviews without being potentially subjected to anchoring bias themselves.

4. **Participant obliviousness to true purpose of study**. Since anchoring would usually be unnoticed by reviewers themselves, it is important to replicate this condition in the experiment. Informing participants of the true purpose of the study could potentially change their behavior according to the demand characteristics effect [50]. In our experiment, we need to conceal the purpose of the study and make it such that participants do not suspect that the study concerns reviewer anchoring.

Addressing challenge 1 enables us to measure an anchoring effect if it exists, while addressing challenges 2–4 ensure that in the absence of an anchoring effect, the ratings received from the control and experimental groups should be equivalent.

These challenges are very tricky to simultaneously address. For example, consider a simple experimental design in which reviewers are randomly assigned to either a high-quality or a low-quality version of a paper; then, after the reviews, experimenters construct a rebuttal to address the points raised in the review. The criticisms raised by the reviewers could concern naturally subjective topics such as its significance. In these cases, we would not be able to refute the reviewer with an objective response in the rebuttal and would struggle to distinguish reviewer anchoring from genuine subjective beliefs (challenge 1). Since the errors in the low-quality paper are due to mistakes by the authors, we would not be able to distinguish between reviewers exhibiting anchoring and reviewers penalizing the author mistakes (challenge 2). Even for the same version of the paper, the criticisms raised by reviewers will likely be widely varied in topic. Thus, if the same rebuttals are used for all reviews, the rebuttals may not match the concerns in each review (challenge 1). Alternatively, if the experimenter generates individualized rebuttals for each review, we cannot guarantee that the post-rebuttal version of the low-quality paper has equivalent quality to the high-quality paper (challenge 3). Finally, if the experiment places significant focus on the rebuttal, participants may suspect the true purpose of the study and modify their behavior accordingly (challenge 4).

**3.1.2 Experimental procedure.** In this subsection, we present our experimental procedure, which addresses each of the aforementioned challenges.

*3.1.2.1 Experimental setting.* The experiment procedure consists of a 30-minute, 1-on-1 Zoom meeting with each participant. Each participant takes the role of a reviewer for one paper within a simulated peer review process, and all participants review the same paper. A snapshot of the paper reviewed is provided in Fig 2. Participants are falsely told that the purpose of the study is to analyze the effect of new types of media (such as animations) on reviews, and are informed that the paper should be reviewed as a submission to an application-focused track of a large AI conference. Participants are given a reviewer form constructed based on the reviewer guidelines in the AAAI 2020 [51] and NeurIPS 2022 [52] conferences. The reviewer form contains scores in five sub-categories {Significance, Novelty, Soundness, Evaluation, Clarity}, one sentence justifications for these scores, as well as an Overall score and a confidence rating. Following the fictitious purpose of the study, the form also asked participants to

# Multimodal Validity Protection for Product Descriptions in E-commerce

**#deployed-applications-of-ai #deep-learning #e-commerce**
**Anonymous Authors**

**Abstract**

In this paper, we introduce Multimodal Validity Protection (MVP), a tool that we developed and deployed on e-commerce websites to flag untrustworthy products. E-commerce platforms currently face an issue of scalability in using human reviewers for approving new products, while recent advances in multimodal machine learning have enabled models to achieve much higher performance in the image captioning task.

**Fig 2. A snapshot of the constructed paper reviewed by participants.** The paper is hosted online and viewed through the participant's browser, allowing for the natural use of an animated GIF figure.

"Please comment on the use of animated figures. (If you did not see this form of media, please answer 'N/A')". This question regarding animated figures plays an important part in our experimental intervention, which we detail in the following paragraph. After the review, we also record participants' institution, program, and year of study.

*3.1.2.2 Intervention.* The key difference between the conditions lies in the presentation of the main evaluation result of the paper. In the control group, this result is presented as an animated GIF graphic (shown in Fig 1A), whereas the experimental group is initially presented a broken version of the GIF that is stuck on the first frame (Fig 1B), which shows a significantly weaker result. Then, when experimental group participants are asked the aforementioned question to comment on animated figures, they would indicate that they had not seen any by answering 'N/A'. After these participants submit their reviews, the experimenter deceives them by saying that their answer was unexpected and that they should have seen an animated figure. In parallel, the experimenter secretly changes the contents of the webpage displaying the paper such that all new visits see the animated GIF in the paper working properly. The experimenter then suggests the participants to refresh the website, upon which the animation loads and they are asked to revise their scores and comments accordingly.

We performed a pilot study with 14 participants before full deployment to test for feasibility and practice the deception. For more details on the deception and score revision process, as well as how deviations from the procedure due to unexpected participant behavior were handled, we refer the reader to S1 Appendix. All of the instructions, interfaces, and the paper contents are available at https://github.com/theryanl/ReviewerAnchoring.

**3.1.3 Design justification.** We now highlight some key aspects of our experimental design and how they address the aforementioned challenges.

- **Construction of the reviewed paper**. In order to ensure that the change in quality between the initial and revised versions of the paper was clear and objective (challenge 1), we manually constructed a single paper for all participants to review. The initial and revised versions differed in the paper's numerical results, as this was an area where the paper's quality could be changed objectively. To make the change in quality clearer, the results between the initial and revised/control versions of the paper were very different, and the paper was constructed to emphasize this result. Additionally, we made the paper heavily application-focused and made its metrics easily interpretable such that our participants (who were at minimum computer science PhD students) would not need any specific technical background to interpret the results.

- **Technical error in displaying the GIF**. In the experimental group, the issue in the initial version of the paper was presented as the result of a technical error (the frozen GIF). Since the error was clearly not attributable to the authors, reviewers could not reasonably justify reflecting the error in their scores, which allowed us to circumvent the author mistake confounder (challenge 2). Additionally, the frozen GIF issue in the initial paper could be corrected for all participants regardless of the specifics of their review. Thus, we were able to ensure that the change seen by the experimental group was both relevant and identical across participants (challenge 1), while the changed paper was also equal to the paper reviewed by the control group (challenge 3).

- **Deceptive experimental purpose**. We created the alternate experimental purpose, "To study the effect of new types of media on reviews", to accomplish three objectives. First, we were able to justify the perceived experimental procedure without mentioning anchoring to participants (challenge 4). Second, we enabled the natural use of animated GIFs in the paper, while not raising suspicion in the case where no GIF was seen. Third, we were able to

naturally include the question asking for comments on the use of animated figures. On one hand, this enabled the experimenter to easily convince participants that there was a technical error by citing their answer. On the other hand, it allowed for the experimenter to naturally ask the participant to refresh the page, allowing the change in the paper to be shown immediately after. Participants were debriefed about the deception and true purpose of the experiment immediately after the study.

## 3.2 Participation and data collection

We recruited 108 participants, who were separated at random into control and experimental groups and were unaware of their assignment. Participants were either PhD students or PhDs with at least one publication in a computer science-related field in the last 5 years (see Table 1). Participants were recruited across nine research universities in the United States through various methods including physical posters, university mailing lists, and social media posts (see S3 Appendix). We conducted a power analysis to determine the target number of participants (see S2 Appendix). As a large fraction of reviewers in computer science conferences are PhD students (e.g., 33% in the NeurIPS 2016 conference [4]), our participant pool is fairly representative of the conference reviewer population we aim to study.

For each participant, we gathered the following data:

1. Overall scores on a 1–10 scale.

2. Category scores in {Significance, Novelty, Soundness, Evaluation, Clarity} on a 1–4 scale and 1-sentence comments justifying each.

3. Confidence in their evaluation on a 1–5 scale.

4. Comments on the hyperlinks and animated figures.

5. Participant-specific information: Institution, program and (if PhD student) year.

The score categories and scales were modeled after those of NeurIPS and AAAI, two of the largest annual computer science conferences. In the experimental group, participants were given a chance to revise all review information after seeing the figure change. In this case, both initial and revised versions were recorded. This resulted in the collection of 3 different sets of data: scores from the control group, initial scores from the experimental group, and revised scores from the experimental group.

After the study, we asked participants a few questions to determine the effectiveness of the deception and ensure that they were oblivious to the true study purpose (i.e., challenge 4 in Section 3.1.1). Before debriefing participants, we asked them if they suspected that the study featured deception; if they answered affirmatively, we asked them to describe what they believed the true study purpose was. If they were able to detect that we deceived them on the study purpose and specifically identify that the true purpose was about re-reviewing or rebuttals, we would exclude them from the study. Along with this, we also included two trivial exclusion criteria: (i) If participants do not consent to their data being collected for the true study

**Table 1. Distribution of participant years of study.**

| | Year of PhD studies | | | | | | Post-PhD |
|---|---|---|---|---|---|---|---|
| | 1st | 2nd | 3rd | 4th | 5th | 6th+ | |
| # Participants | 17 | 28 | 18 | 20 | 12 | 6 | 7 |

purpose, and (ii) if participants do not finish the study. For reference, participants were compensated $20 for participation in the study, and were allowed to withdraw at any time for partial ($10-$15) compensation. No participants withdrew or were excluded due to these criteria (or for any other reason), demonstrating the effectiveness of the deception in our experiment design.

### 3.3 Analysis

We first performed a preliminary test of the validity of our experimental setup by comparing the initial scores $I$ provided by the experimental group with the the scores $C$ provided by the control group. If our experimental setup was successful at inducing a perceived difference in paper quality, we should see that the initial experimental scores are generally lower than the control scores. To compare the distributions of these scores, we performed a non-parametric test of the null hypothesis that the control and initial scores have the same distribution. Specifically, we conducted a one-sided permutation test (with 100000 permutations) with the Mann-Whitney $U$ statistic against the alternative hypothesis that the distribution of the control scores $C$ is stochastically greater than the distribution of the initial scores $I$. The test statistic is

$$U = \sum_{C_i \in C} \sum_{I_j \in I} S(C_i, I_j), \qquad \text{where } S(a, b) = \begin{cases} 1 & \text{if } a > b \\ 1/2 & \text{if } a = b \, . \\ 0 & \text{if } a < b \end{cases} \tag{1}$$

We performed two tests between these groups, comparing both the Overall scores and the Evaluation category scores. We chose to analyze the Evaluation category, defined as "a score for how its evidence supports its conclusions [. . .]", as we expected our experimental manipulation to have the greatest effect in this category. Across the two tests, we controlled the false discovery rate using the Benjamini-Hochberg correction under the assumption that the test statistics are positively dependent [53], and the p-values we report are adjusted for this correction [54]. As effect sizes, we also report point estimates of the Vargha-Delaney $A$ statistic [16], computed as $A = \frac{U}{|C||I|}$, along with 95% bootstrapped confidence intervals (using 100000 samples).

Our primary analysis aims to detect anchoring in reviewers. To test for the anchoring effect, we compared the revised scores $R$ provided by the experimental group with the scores $C$ provided by the control group. For this, we performed a non-parametric test of the null hypothesis that the control and revised scores have the same distribution. We again used a one-sided permutation test with the Mann-Whitney $U$ statistic against the alternative hypothesis that the distribution of the control scores is stochastically greater than the distribution of the revised experimental scores. The test statistic is

$$U = \sum_{C_i \in C} \sum_{R_j \in R} S(C_i, R_j). \tag{2}$$

We performed two tests to compare both the Overall scores and the Evaluation category scores, and again controlled the false discovery rate at $\alpha = 0.05$ across the two tests using the Benjamini-Hochberg correction (again assuming positive dependence). We report the Vargha-Delaney $A$ statistic as the effect size, with estimates computed as $A = \frac{U}{|C||R|}$.

As stated in Section 3, our research question and study design were pre-registered. However, the analysis specified here differs from the analysis plan specified in the preregistration. In the preregistration, the test statistic was specified to be the difference between the mean

scores of each group, and only the Overall scores were to be analyzed. However, as the scores are not necessarily on a linear scale (in fact, they were each given a description on the review form), the arithmetic means of the scores are not as meaningful. We also analyzed Evaluation category scores since our experimental design specifically manipulates the paper quality in this category. The tests of the validity of our experimental setup were also not preregistered. The preregistered original analysis is available at https://aspredicted.org/W94_GD3.

Code for all analyses is provided at https://github.com/theryanl/ReviewerAnchoring.

## 4 Results

### 4.1 Main results

The results of our main hypothesis tests introduced in Section 3.3 are reported in Table 2.

Our comparisons between the initial scores and control scores to test the validity of our experimental setup resulted in effect sizes $A = 0.5857$ with respect to the Overall scores (adjusted $p = 0.0575$) and $A = 0.6375$ with respect to the Evaluation category scores (adjusted $p = 0.0096$). The effect sizes can be interpreted as the probability that a randomly chosen control score is greater than a randomly chosen initial score, breaking ties uniformly at random. An effect size of $A = 0.5$ means that the two distributions are stochastically similar, and higher values of $A$ indicate the extent to which the distribution of control scores is stochastically greater. If our experimental setup successfully created a perceived difference in paper quality between the conditions, we expect the control scores to be higher than the initial scores (corresponding to effect sizes $A > 0.5$). While both comparisons had moderate effect sizes, the comparison in the Evaluation category is significant at $\alpha = 0.01$, while the comparison in Overall scores is significant at $\alpha = 0.1$. This provides evidence that the paper quality was perceived as different between the two groups, although reviewers may not have reflected this difference as much in their Overall scores.

Given that our experiment successfully constructed an environment where anchoring could occur, we turn to our analysis of whether anchoring did occur. Our comparisons between the revised scores and control scores, which test for the anchoring effect, resulted in effect sizes $A = 0.5048$ with respect to the Overall scores (adjusted $p = 0.6064$) and $A = 0.4863$ with respect to the Evaluation category scores (adjusted $p = 0.6064$). Recall that in the presence of an anchoring effect, we expect the control scores to be higher than the revised scores (corresponding to effect sizes $A > 0.5$). Both statistics are insignificant at $\alpha = 0.1$ (and would have been insignificant even without Benjamini-Hochberg correction), indicating that our analysis failed to reject the null hypothesis that reviewers do not anchor. In other words, we did not find any evidence of anchoring bias.

**Table 2. Results of comparisons between initial or revised scores from the experimental group and scores from the control group, with respect to both Overall and Evaluation scores.**

| Experimental Condition | Score Type | *A* | 95% Confidence Interval | p-value | Experimental Condition Mean | Control Mean |
|---|---|---|---|---|---|---|
| Initial | Overall | 0.5857 | [0.4793, 0.6881] | 0.0575 | 5.519 | 6.037 |
| Initial | Evaluation | 0.6375 | [0.5381, 0.7327] | 0.0096 | 1.908 | 2.352 |
| Revised | Overall | 0.5048 | [0.3981, 0.6109] | 0.6064 | 5.907 | 6.037 |
| Revised | Evaluation | 0.4863 | [0.3836, 0.5888] | 0.6064 | 2.389 | 2.352 |

The effect size $A$ is the Vargha-Delaney $A$ statistic, with 95% confidence intervals constructed via bootstrap. Benjamini-Hochberg adjusted p-values are reported. The rightmost two columns show the mean Overall (1–10 scale) or Evaluation (1–4 scale) scores within the experimental group (initial or revised score) and the control group.

## 4.2 Supplemental results

In addition to the main test statistic, we also performed the following informal supplemental analyses. As these analyses were exploratory and data-dependent, the observations we made in these analyses should be interpreted primarily as motivation for future work and not as support for statistically significant conclusions.

**4.2.1 Other category scores.**   In Table 3, we show the results of additional comparisons conducted between revised scores and control scores. We compared scores for each of the categories on the review form apart from the Evaluation category analyzed earlier. We used the same methodology as in our main analysis to compute the effect sizes and 95% confidence intervals. Overall, these results do not indicate that other categories showed signs of anchoring.

**4.2.2 Confidence.**   We additionally conducted comparisons to investigate whether anchoring was associated with the self-reported confidence of reviewers. In Table 4, we separate participants into two groups based on their self-reported confidence score, given on a scale of of 1–5: *confident*, where participants reported a score of 3 ("Fairly Confident") or higher, and *unconfident* where they reported a score of 2 ("Willing to defend") or lower. This threshold between confident and unconfident reviewers was chosen before the analysis based on the stated descriptions of the scores. In both the control and experimental groups, there were 41 confident reviewers and 13 unconfident reviewers. We conducted comparisons between the revised Overall scores from the experimental group and the Overall scores from the control group, and found that confident reviewers had the same mean revised and control scores, while unconfident reviewers had generally lower revised scores (indicated by the $A = 0.63$ effect size). This could indicate that unconfident reviewers are more likely to exhibit anchoring. However, since there were less unconfident reviewers, the uncertainty around this effect size is large.

**4.2.3 Seniority.**   Next, we split participants into less experienced ("junior") and more experienced ("senior") reviewers, and conducted a comparison between the revised and control Overall scores for each subgroup in Table 5. Junior reviewers were PhD year 3 and under, whereas senior reviewers were PhD year 4 and over or beyond their PhD. This threshold was

**Table 3. Results of comparisons between revised and control scores in the remaining categories.**

| Category | A | 95% Confidence Interval | Revised Mean | Control Mean |
|---|---|---|---|---|
| Significance | 0.5065 | [0.4119, 0.6010] | 2.78 | 2.83 |
| Novelty | 0.4746 | [0.3745, 0.5736] | 2.48 | 2.46 |
| Soundness | 0.4863 | [0.3849, 0.5898] | 2.76 | 2.69 |
| Clarity | 0.4609 | [0.3614, 0.5621] | 3.31 | 3.17 |

The effect size A is the Vargha-Delaney A statistic, with 95% confidence intervals constructed via bootstrap. The rightmost two columns show the mean scores (1–4 scale).

**Table 4. Results of comparisons between revised and control Overall scores for both confident (3+, 1–5 scale) and unconfident (2-) reviewers.**

| Subgroup | # Experimental | # Control | A | 95% Confidence Interval | Revised Mean | Control Mean |
|---|---|---|---|---|---|---|
| Confident | 41 | 41 | 0.4685 | [0.3453, 0.5925] | 6.00 | 6.00 |
| Unconfident | 13 | 13 | 0.6331 | [0.4201, 0.8284] | 5.62 | 6.15 |

The effect size A is the Vargha-Delaney A statistic, with 95% confidence intervals constructed via bootstrap. The rightmost columns show the mean scores (1–10 scale).

**Table 5. Results of comparisons between revised and control Overall scores, for both junior (PhD years 1–3) and senior (4+) reviewers.**

| Subgroup | # Experimental | # Control | A | 95% Confidence Interval | Revised Mean | Control Mean |
|---|---|---|---|---|---|---|
| Junior | 26 | 37 | 0.4865 | [0.3415, 0.6284] | 5.96 | 6.00 |
| Senior | 28 | 17 | 0.5357 | [0.3739, 0.6975] | 5.86 | 6.12 |

The effect size *A* is the Vargha-Delaney *A*, with 95% confidence intervals constructed via bootstrap. The rightmost two columns show the mean scores (1–10 scale).

chosen before the analysis to produce the most equally-sized groups. We found similar results across the two subgroups, suggesting that our study results may not be dependent on the large amount of junior participants we have in comparison to real conference settings, though the uncertainty around the effect size is large.

**4.2.4 Counts of score changes.** Though our main analysis did not find evidence of anchoring (as shown in Section 4.1), we observe that, consistent with the findings from previous conference organizers in Section 2.2, a majority of the reviewers in the experimental group did not change their given scores (see Table 6). Out of 54 experimental group participants, 15 (28%) changed their Overall score, with nine participants raising their Overall scores by 1 and six raising their Overall scores by 2. Meanwhile, 25 (46%) participants changed one or more category scores, with 22 (41%) participants including a change in the Evaluation category. Other category scores were changed by only a few participants, which was expected as our manipulation primarily targeted the Evaluation category. In Table 7, we further break down the scores and comments updated by experimental group participants.

# 5 Conclusion and discussion

In this paper, we presented the design and results of a randomized controlled experiment to test for reviewer anchoring bias in conference peer review. Our design carefully addresses

**Table 6. Breakdown of experimental group participants by those who changed either their Overall score or their score in at least one category.**

| | Overall score unchanged | Overall score changed | Total |
|---|---|---|---|
| Category scores unchanged | 28 | 1 | 29 |
| Category scores changed | 11 | 14 | 25 |
| Total | 39 | 15 | 54 |

Most (> 50%) participants changed neither their category nor their Overall scores.

**Table 7. Number of experimental group participants (out of 54 total) who changed their scores or comments in each category and Overall.**

| Category | # Participant scores changed | | # Participant comments changed | |
|---|---|---|---|---|
| Significance | 7 | (13%) | 9 | (17%) |
| Novelty | 1 | (2%) | 0 | (0%) |
| Soundness | 6 | (11%) | 5 | (9%) |
| Evaluation | 22 | (41%) | 31 | (57%) |
| Clarity | 0 | (0%) | 2 | (4%) |
| Overall | 15 | (28%) | – | – |

Due to timing constraints, comments on the Overall score were not collected.

various challenges and confounders through the employment of animated media, deception, and an overarching cover story.

Our main analysis did not find evidence of the existence of reviewer anchoring effects in peer review. In the absence of anchoring, the lack of change in scores and decisions observed in conference rebuttal phases may be due to other reasons, such as rebuttals having a relatively weak impact on the quality of the paper, or reviewers penalizing the paper for statements that were unclear or misunderstood in the initial submitted version. Another significant issue concerning the rebuttal process is the limited participation from reviewers [1, 4]. Regardless of the prevalence of anchoring, it is essential for conferences to address this lack of active participation in the review processes.

Our study had several limitations which we now discuss. One potential limitation was that our sample size could have resulted in insufficient statistical power to detect an effect. Although we estimated the sample size needed for our experiment using real conference data (see S2 Appendix), the variance in the collected scores was higher than that of the data we used. This variance in scores could have been due to the lack of a unifying context or set of norms that conference reviewers in the same subfield would have. Thus, future studies can consider recruiting participants with expertise in one particular subfield to help increase the calibration between reviewers.

Another possibility is that, even if anchoring is prevalent in real conference settings, the experimental conditions of our study failed to replicate the conference environment sufficiently to induce this same effect. For example, a common piece of feedback we received from participants in the study was that there was no context behind the result in the paper. Some participants expressed uncertainty in their review as to whether the weak initial result is significant, and retained this even for the larger corrected result. In contrast, reviewers in a real conference may have better knowledge to more accurately judge the significance of a paper's contributions. In future studies, the aspects of the paper that are updated during the rebuttal may need to be more clearly interpretable to the entire study population, which could also be resolved by recruiting participants with expertise in a particular subfield.

Additionally, our experiment intentionally omits certain elements that are typically present in a real conference environment, some of which may be responsible for reviewer anchoring in the real setting. One such aspect is the social dynamic of reviewers. For example, if reviewers know that other reviewers and area chairs can observe their reviews, it is possible that they would choose to defend their initial position more due to concerns about their image in front of others. Similar social dynamics may be present when reviewers are asked to engage directly with authors in discussions. However, the social aspect may also introduce various confounding effects such as reviewers being influenced by the scores of other reviews [46]. We decided to forgo the capturing of these secondary social effects, instead leaving them to future work.

Another limitation of our work is that we run our experiment with only one paper, which could lead to our findings to be less generalizable. There is precedence of research involving reviewers reviewing fake papers, and in each of these only 1 to 3 papers are constructed [55–59]. Due to the high sample size determined from the power analysis (see S2 Appendix) and the limited pool of eligible participants (see S3 Appendix), we chose to have one paper to reduce the sample size needed in order to test for statistical significance, as having multiple papers would require an additional random effect to be modeled. Future work may also include papers from multiple domains to bolster the generalizability of the study.

Finally, there are other variations of our research question that future work could consider. Our supplemental analysis with respect to reviewer confidence suggests that the answer to our research question may not be homogeneous across the entire reviewer pool. Future work may

want to design experiments that more carefully take this consideration into account by testing for effects within subpopulations.

## Supporting information

**S1 Appendix. Deception and revision process.**
(PDF)

**S2 Appendix. Power analysis.**
(PDF)

**S3 Appendix. Participant recruitment.**
(PDF)

## Author Contributions

**Conceptualization:** Ryan Liu, Steven Jecmen, Vincent Conitzer, Fei Fang, Nihar B. Shah.

**Data curation:** Ryan Liu, Nihar B. Shah.

**Formal analysis:** Ryan Liu, Steven Jecmen.

**Funding acquisition:** Vincent Conitzer, Fei Fang, Nihar B. Shah.

**Investigation:** Ryan Liu.

**Methodology:** Ryan Liu, Steven Jecmen, Vincent Conitzer, Fei Fang, Nihar B. Shah.

**Project administration:** Nihar B. Shah.

**Resources:** Ryan Liu.

**Software:** Ryan Liu.

**Supervision:** Vincent Conitzer, Fei Fang, Nihar B. Shah.

**Visualization:** Ryan Liu, Steven Jecmen.

**Writing – original draft:** Ryan Liu, Steven Jecmen.

**Writing – review & editing:** Ryan Liu, Steven Jecmen, Fei Fang, Nihar B. Shah.

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
