## [Decision Letter · Decision Letter 0]

16 Oct 2023

PONE-D-23-21814Testing for Reviewer Anchoring in Peer Review: A Randomized Controlled TrialPLOS ONE

Dear Dr. Liu,

Thank you for submitting your manuscript to PLOS ONE. After careful consideration, we feel that it has merit but does not fully meet PLOS ONE’s publication criteria as it currently stands. Therefore, we invite you to submit a revised version of the manuscript that addresses the points raised during the review process.

We look forward to receiving your revised manuscript.

Kind regards,

Stephan Leitner

Academic Editor

PLOS ONE

“This work was supported in part by NSF CAREER Award 1942124 and NSF 2200410.”

“NS received the National Science Foundation CAREER Award 1942124 (https://www.nsf.gov/). The funders had no role in study design, data collection and analysis, decision to publish, or preparation of the manuscript.

NS and FF received the National Science Foundation Communications and Information Foundations 2200410 (https://new.nsf.gov/funding/opportunities/ccf-communications-information-foundations-cif). The funders had no role in study design, data collection and analysis, decision to publish, or preparation of the manuscript.”

4. We note that Figure 2 in your submission contain copyrighted images. All PLOS content is published under the Creative Commons Attribution License (CC BY 4.0), which means that the manuscript, images, and Supporting Information files will be freely available online, and any third party is permitted to access, download, copy, distribute, and use these materials in any way, even commercially, with proper attribution. For more information, see our copyright guidelines: http://journals.plos.org/plosone/s/licenses-and-copyright.

Additional Editor Comments:

Dear authors,

We have received two reviews for your paper from expert reviewers. While one is very positive, the other is slightly more critical. I would like to offer you the chance to address these issues in a revision, particularly by extending the discussion of results and limitations as suggested by the more critical reviewer.

Regards,

Stephan Leitner

Reviewers' comments:

Reviewer's Responses to Questions

**Comments to the Author**

1. Is the manuscript technically sound, and do the data support the conclusions?

Reviewer #1: Yes

Reviewer #2: Partly

2. Has the statistical analysis been performed appropriately and rigorously? 

Reviewer #1: Yes

Reviewer #2: Yes

3. Have the authors made all data underlying the findings in their manuscript fully available?

Reviewer #1: Yes

Reviewer #2: Yes

4. Is the manuscript presented in an intelligible fashion and written in standard English?

Reviewer #1: Yes

Reviewer #2: Yes

5. Review Comments to the Author

Reviewer #1: Thanks for this nicely designed and conducted experiment. The paper is innovative and the mild deception you use tolerable. I like the paper and results and thus support publication. Here just a few minor points you may change in the paper:

- page 1, second line from bottom: replace "change" with "adapt"

- page 11, line under Section 4 - delte this unnecessary sentence "In this section..."

Reviewer #2: In this paper, the authors investigate an intriguing phenomenon in peer reviews, known as "Anchoring", where reviewers simply do not adjust their scores as much as they should. Through a randomized controlled trial, the paper studies the anchoring effect in peer reviews. The experiment designed by the authors involved 108 researchers with varied research backgrounds. The main findings of the authors include: 1) paper quality was perceived differently between the two groups; 2) no evidence of anchoring bias was found.

The paper is meticulously designed, addressing some challenges, which showcases the substantial effort put forth by the authors. The writing is clear, the results are presented succinctly, and the overall presentation is good. Additionally, the authors have engaged in a degree of discussion regarding the limitations of their method. Overall, the article makes a certain contribution to anchoring in peer review.

Pros:

1. The research questions are interesting and novel.

2. The experiments are well designed. The experiment design of the article is commendable and incurred certain costs, including the recruitment of trial participants and data collection, etc.

3. The writing of the paper is very good, including sections like the abstract, introduction, and conclusion. The authors have highlighted their main conclusions and also discussed other findings like the involvement of junior participants.

Cons:

1. The experimental setup of the article has certain limitations, such as the construction of only one fake paper.

2. The data provided by the authors is somewhat scant, although they have discussed the limitations.

3. The gap between the authors' experiments and the real world is considerable, making it challenging to assure the validity of the findings on real-world data.

My primary concerns regarding the article are as follows:

As mentioned in the paper, the experimental design is overly idealized and deviates significantly from real-world scenarios. Firstly, in real-world situations, especially in computer conference papers, a reviewer typically reviews multiple papers, and a paper receives reviews from multiple reviewers. This introduces elements of peer effect and peer pressure, which are considered crucial factors [1]. Furthermore, the revision of scores by reviewers may also depend on the authors' rebuttal skills or rebuttal politeness [2,3,4].

These aspects are not reflected in the experiments conducted in the paper. While considering these factors is challenging, if the authors claim to study the anchoring phenomenon in peer reviews, then these factors cannot be overlooked.

Other minor suggestions:

Please revise some arxiv paper reference into their proceedings versions.

References:

[1] Gao, Yang, et al. "Does My Rebuttal Matter? Insights from a Major NLP Conference." Proceedings of the 2019 Conference of the North American Chapter of the Association for Computational Linguistics: Human Language Technologies, Volume 1 (Long and Short Papers). 2019.

[2] Huang, Junjie, et al. "What makes a successful rebuttal in computer science conferences?: A perspective on social interaction." Journal of Informetrics 17.3 (2023): 101427.

[3] Rogers, Anna, and Isabelle Augenstein. "What Can We Do to Improve Peer Review in NLP?." Findings of the Association for Computational Linguistics: EMNLP 2020. 2020.

[4] Bharti, Prabhat Kumar, et al. "PolitePEER: does peer review hurt? A dataset to gauge politeness intensity in the peer reviews." Language Resources and Evaluation (2023): 1-23.

6. PLOS authors have the option to publish the peer review history of their article (what does this mean?). If published, this will include your full peer review and any attached files.

Reviewer #1: No

Reviewer #2: No

---

## [Author Response · Author response to Decision Letter 0]

2 Mar 2024

Reviewer #1: 

Thank you for your careful review of our manuscript. We have made changes to the paper according to your following suggestions: 

- page 1, second line from bottom: replace "change" with "adapt"

- page 11, line under Section 4 - delte this unnecessary sentence "In this section..."

You can find these changes in the “track changes” version of the revision.

Reviewer #2: 

Thank you for your detailed comments and suggestions for our manuscript. In particular, we appreciate your careful consideration the various facets of our experiment design. We would like to briefly discuss your concerns regarding our paper:

1. The experimental setup of the article has certain limitations, such as the construction of only one fake paper.

Response: 

We agree with the reviewer that this is indeed a limitation to our work. We have added a paragraph in the discussion specifically discussing this limitation. We reproduce the paragraph here for the reviewer's convenience: 

“Another limitation of our work is that we run our experiment with only one paper, which could lead to our findings to be less generalizable. There is precedence of research involving reviewers reviewing fake papers, and in each of these only 1 to 3 papers are constructed [1–5]. Due to the high sample size determined from the power analysis (see S2 Appendix) and the limited pool of eligible participants (see S3 Appendix), we chose to have one paper to reduce the sample size needed in order to test for statistical significance, as having multiple papers would require an additional random effect to be modeled. Future work may also include papers from multiple domains to bolster the generalizability of the study.” 

2. The data provided by the authors is somewhat scant, although they have discussed the limitations.

Response:

We provide the full anonymized numerical responses and participant institution and year in the data folder of https://github.com/theryanl/ReviewerAnchoring. Our IRB approval and the consent of participants do not permit us to include participant comments in the dataset. This was due to potential concerns of anonymity breach (e.g., “Since I work in the neighboring field of xxx, …”). The participant pool, mainly CS or CS-related field PhDs, as well as their institution and year are provided. If additional information regarding their area of study can be inferred from their responses, this may cause certain participants’ identities to be identifiable.

3. The gap between the authors' experiments and the real world is considerable (e.g., peer effect, peer pressure, rebuttal skills, rebuttal politeness), making it challenging to assure the validity of the findings on real-world data. While considering these factors is challenging, if the authors claim to study the anchoring phenomenon in peer reviews, then these factors cannot be overlooked.

Response:

As you mention, there are many other peer effects in the process of rebuttals. The design of this experiment deliberately avoids adding these additional effects in order to isolate the effect we wish to understand – anchoring in its base form (as defined by [6]) without social or author effects. We believe this base form is important to analyze, as an existence of anchoring in this setting would mean that the re-reviewing paradigm itself causes anchoring bias to happen, meaning that altering social pressures and other effects would be insufficient to avoid bias. This setting is also consistent with anchoring bias studies in psychology which did not contain external social pressures or author skills or politeness [6]. 

In this case, adding other peer effects or author skills and politeness can confound any observations that we make. These factors are orthogonal to anchoring, and our objective in the experiment design is to single out anchoring and avoid these other confounders. This is actually a flaw with the study by Gao et al. you cited – the paper claims "peer pressure" as a reason for their observation, but their study does not isolate such effects and hence the claim can be confounded by various factors such as program chair instructions to arrive at a consensus. Our study, in contrast, asks a specific question (of anchoring), and the experiment design deliberately and carefully avoids other such confounders in order to answer this specific question. By coming to a conclusion on just anchoring, this also allows us to better evaluate the existence of other effects in future studies, and possibly help inform the design of future peer review paradigms. 

4. Please revise some arxiv paper reference into their proceedings versions.

As you suggested, we have made changes to the references, replacing arxiv paper references to their proceedings versions. They should be up-to-date with Google Scholar as of 2/9/2024. 

[1] W. G. Baxt et al. “Who reviews the reviewers? Feasibility of using a fictitious manuscript to evaluate peer reviewer performance”. In: Annals of emergency medicine (1998).

[2] G. B. Emerson et al. “Testing for the presence of positive-outcome bias in peer review: a randomized controlled trial”. In: Archives of internal medicine (2010).

[3] F. Godlee, C. R. Gale, and C. N. Martyn. “Effect on the quality of peer review of blinding reviewers and asking them to sign their reports: a randomized controlled trial”. In: JAMA (1998).

[4] S. Schroter et al. “Effects of training on quality of peer review: randomised controlled trial”. In: BMJ (2004).

[5] S. Schroter et al. “What errors do peer reviewers detect, and does training improve their ability to detect them?” In: Journal of the Royal Society of Medicine (2008).

[6] A. Tversky and D. Kahneman. “Judgment under uncertainty: Heuristics and biases.” In: Science (1974).

---

## [Decision Letter · Decision Letter 1]

12 Mar 2024

Testing for Reviewer Anchoring in Peer Review: A Randomized Controlled Trial

PONE-D-23-21814R1

Dear Dr. Liu,

We’re pleased to inform you that your manuscript has been judged scientifically suitable for publication and will be formally accepted for publication once it meets all outstanding technical requirements.

Kind regards,

Stephan Leitner

Academic Editor

PLOS ONE

Additional Editor Comments (optional):

We have now received two reviews from the experts who evaluated your manuscript in the previous round. I am very pleased to inform you that, based on the reviewers' assessments and my own evaluation of your manuscript, we can now accept it for publication.

Reviewers' comments:

Reviewer's Responses to Questions

**Comments to the Author**

1. If the authors have adequately addressed your comments raised in a previous round of review and you feel that this manuscript is now acceptable for publication, you may indicate that here to bypass the “Comments to the Author” section, enter your conflict of interest statement in the “Confidential to Editor” section, and submit your "Accept" recommendation.

Reviewer #1: All comments have been addressed

Reviewer #2: All comments have been addressed

2. Is the manuscript technically sound, and do the data support the conclusions?

Reviewer #1: Yes

Reviewer #2: Yes

3. Has the statistical analysis been performed appropriately and rigorously? 

Reviewer #1: Yes

Reviewer #2: Yes

4. Have the authors made all data underlying the findings in their manuscript fully available?

Reviewer #1: Yes

Reviewer #2: Yes

5. Is the manuscript presented in an intelligible fashion and written in standard English?

Reviewer #1: Yes

Reviewer #2: Yes

6. Review Comments to the Author

Reviewer #1: (No Response)

Reviewer #2: Thanks to the author for the reply, although I still think that peer pressure must be considered when analyzing the anchoring effect of peer review (peer review opinions are obviously given by multiple people together, and few papers are reviewed by only one reviewer).

The reference [6] is not the definition of anchoring effect in peer review. However, I agree that the author used randomized experiments to verify the anchoring effect.

While PLOS ONE does not attempt to use the peer review process to determine whether or not an article reaches the level of 'importance' required by a given journal, PLOS ONE uses peer review to determine whether a paper is technically rigorous and meets the scientific and ethical standards for inclusion in the published scientific record.

So I am OK with the current revisions.

7. PLOS authors have the option to publish the peer review history of their article (what does this mean?). If published, this will include your full peer review and any attached files.

Reviewer #1: No

Reviewer #2: No

---

## [Editor Report · Acceptance letter]

17 Jul 2024

PONE-D-23-21814R1 

PLOS ONE

Dear Dr. Shah, 

I'm pleased to inform you that your manuscript has been deemed suitable for publication in PLOS ONE. Congratulations! Your manuscript is now being handed over to our production team.

Kind regards, 

on behalf of

Dr. Stephan Leitner 

Academic Editor

PLOS ONE